# Multi-View Travel Time Prediction Based on Electronic Toll Collection Data

**DOI:** 10.3390/e24081050

**Published:** 2022-07-30

**Authors:** Sijie Luo, Fumin Zou, Cheng Zhang, Junshan Tian, Feng Guo, Lyuchao Liao

**Affiliations:** 1Fujian Key Laboratory for Automotive Electronics and Electric Drive, Fujian University of Technology, Fuzhou 350118, China; sjluo@fjut.edu.cn (S.L.); 2191901003@smail.fjut.edu.cn (J.T.); 2College of Computer and Data Science, Fuzhou University, Fuzhou 350108, China; n180310004@fzu.edu.cn; 3College of Information Technology and Management, Hunan University of Finance and Economics, Changsha 410205, China; 201805230240@mails.hufe.edu.cn; 4Fujian Provincial Big Data Research Institute of Intelligent Transportation, Fujian University of Technology, Fuzhou 350118, China; achao@fjut.edu.cn

**Keywords:** expressway, electronic toll collection, travel time, vehicle type, spatial proximity

## Abstract

The travel time prediction of vehicles is an important part of intelligent expressways. It can not only provide the vehicle distribution trend of each section for the expressway management department to assist the fine management of the expressway, but it can also provide owners with dynamic and accurate travel time prediction services to assist the owners to formulate more reasonable travel plans. However, there are still some problems in the current travel time prediction research (e.g., different types of vehicles are not processed separately, the proximity of the road network is not considered, and the capture of important information in the spatial-temporal perspective is not considered in depth). In this paper, we propose a Multi-View Travel Time Prediction (MVPPT) model. First, the travel times of different types of vehicles of each section in the expressway are analyzed, and the main differences in the travel times of different types of vehicles are obtained. Second, multiple travel time features are constructed, which include a novel spatial proximity feature. On this basis, we use CNN to capture the spatial correlation and the spatial attention mechanism to capture key information, the BiLSTM to capture the time correlation of time series, and the time attention mechanism capture key time information. Experiments on large-scale real traffic data demonstrate the effectiveness of our proposal over state-of-the-art methods.

## 1. Introduction

As an important part of transportation infrastructure, expressways provide important support for social economic development and people’s quality of life. However, in recent years, the number of vehicles in China has gradually increased, and the management and planning of expressways have faced many problems. In order to improve the efficiency of vehicle management of expressways, China’s road management departments have deployed more than 20,000 sets of gantry equipment on expressways across China [1], and the intelligent charging and real-time location recording of vehicles have been realized, which further promotes the fine management of expressways [2]. This is of great significance to improve expressway traffic efficiency, reduce logistics costs, facilitate mass travel, and promote the high-quality development of expressways. At the same time, due to the construction of expressway infrastructure, the expressway gantry system also generates massive Electronic Toll Collection (ETC) data, which provides data support for the “ETC plus”. ETC data record most of the vehicles driving on the expressway, which basically reflects the traffic status of the expressway section [3]. Therefore, through ETC data, we can accurately obtain the road utilization rate, traffic rate, traffic speed, travel time, etc., which can help us effectively predict the traffic flow [4], travel time [5], and speed [6] of all vehicle types in each section of the expressway.

Travel time prediction is an important part of an intelligent expressway, which provides travelers with travel time information of each path, helps travelers make more intelligent travel decisions, and formulates more accurate and reasonable expressway driving schedules [7]. In addition, travel time prediction can also provide auxiliary decision-making information for road management and rectification for traffic management departments. A large number of researchers have studied travel time prediction [5,7,8], with the deepening of research, the error value is gradually reduced, but there are still some problems. First, there is no separate discussion of different types of vehicles, and different types of vehicles on expressways have different travel characteristics [8]. Most researchers predict the future travel time based on the overall travel time of all vehicles, which may seriously affect the accurate prediction of travel time and cannot be applied to fine expressway management. Second, the road proximity is not considered, which has great influence on traffic prediction [9]. However, the proximity of the road network is not considered in the current travel time prediction research. Third, the important information extraction does not consider both spatial and temporal features. In the long-term sequence processing, different information of time dimension and space dimension has different weights for the prediction model [10,11]. If more weight can be given to the important information of the two dimensions of time and space, the accuracy of the prediction model could be further improved.

To address the aforementioned challenges, we propose a Multi-View Travel Time Prediction Model (MVTTP). First, the travel time characteristics of each vehicle type on the expressway are analyzed, and the vehicles are classified according to the difference in travel time of each vehicle type. On this basis, we consider multi-view spatial-temporal features to construct feature vectors, where we propose a novel spatial proximity feature. Finally, we propose a new deep learning framework in which the Convolutional Neural Network (CNN) captures the spatial dependency of the network structure and then adds a spatial attention mechanism to weight the important information, and the Bi-directional Long Short-Term Memory (BiLSTM) captures the temporal dependency of the time series and adds a temporal attention mechanism to capture the important temporal features. The experimental results show that the predicted values after classification are closer to the real travel time values and that the model has better prediction performance. The prediction performance of the model is also improved after considering the spatial proximity, and the proposal has better prediction performance compared with other deep learning methods.

Our contributions are summarized below:We analyze the travel time of expressway, find out there are great differences in the travel time of different types of vehicles, and further verify the necessity of separate predictions for different types of vehicles.We propose a road network proximity feature for travel time prediction, which can perceive the correlation of adjacent sections in the space of the road network.We propose a novel travel time prediction model, which considers the road network proximity, temporal and spatial correlation, and can capture the key spatial-temporal information.We conducted extensive experiments on real- traffic datasets. The results show that our method consistently outperforms the competing baselines.

The organization of this paper is as follows: The first section is the introduction; the second section is the related work. The third section is the details in the Methodology. The fourth section is the Experimental Results and Analysis, and finally, the fifth section is the Conclusion.

## 2. Related Work

At present, travel time prediction models can be divided into two categories, one is model driven, the other is data driven [12]. Model-driven methods were a common travel time prediction model in the past [13], which are mainly divided into queuing models and cell transmission models. With the development of business and industry, a large amount of data is collected, and data-driven prediction has become a hot research topic [14], which can be divided into machine learning algorithms and deep learning models.

Model-driven methods predict future travel time by modeling parameters in traffic models [12]. In the development of model-driven travel time prediction, Takaba et al. [15] use leakage a model and delay model based on queuing theory to predict travel time; their results show that the leakage model has a better performance than the delay model. Skabardonis et al. [16] take the free travel time and traffic signal delay time as the total travel time of the vehicle and used the queuing theory model based on motion wave theory to predict total travel time. Juri et al. [17] combined statistical prediction technology with cell transmission model, using a sliding window framework for online travel time prediction, which is a point-to-point, online, short-term prediction method. Seybold et al. [18] proposed an improved cell transmission model to predict travel time, which uses the least square method and global least square method to optimize model parameters. Model-driven prediction has a relatively complete traffic model and theoretical system, which can clearly explain the relationship between various traffic volumes. However, its prediction time is short, and its prediction performance is not good.

The data-driven method mainly uses a large number of historical data to conduct the model learning and parameter optimization. Then, the model can achieve the effect of an approximately real situation [19]. It is mainly divided into the traditional time series prediction method, the machine learning method, and the deep learning model [20]. The traditional time series prediction algorithms mainly include Autoregressive Integrated Moving Average (ARIMA) and Historical Average (HA), which were once widely used in the field of traffic forecasting [21]. However, since these methods are based on historical records for forecasting, they cannot capture the context features of the data and are gradually replaced by machine learning algorithms.

With the development of machine learning and deep learning, a large number of travel time prediction methods based on machine learning or deep learning have been proposed. Before 2016, Machine learning is a hot research topic [22,23], most of the research on travel time prediction was based on machine learning methods and feature vectors for travel time prediction. However, in 2016, the deep learning system AlphaGo developed by Google defeated the championship of human chess, the deep learning once again became a research hotspot in various fields [24,25,26]. Since then, deep learning-based travel time has also become a research hotspot in the field of transportation.

The mainstream algorithms of machine learning for travel time prediction include Support Vector Regression (SVR), K-Nearest Neighbor (KNN), Linear Regression (LR), Adaptive Boosting (AdaBoost), etc. Kwon et al. [7], based on LR, used stepwise variable selection and decision trees to predict the travel time of expressways. Rice et al. [27] also proposed an improved LR with time-varying coefficient, which uses historical time and the traffic condition of the day to predict travel time. Vanajakshi et al. [28] used SVR to predict short-term travel time. Based on this, Qiu et al. [29] predicted travel time by using floating car trajectory data and radar velocity data based on SVR. Castro-Neto et al. [30] proposed an online support vector machine (OL-SVR) for travel time prediction in atypical traffic conditions such as traffic accidents, bad weather, and holidays. Yao et al. [31] used the travel time, traffic flow, and road occupancy of historical time as the input of SVR, and selected Gaussian radial basis function as the kernel function to predict the travel time. Wang et al. [32] predicted the travel time based on the improved KNN, using cross validation to determine the selection of the k value. Yao et al. [33] selected the training feature and the most similar neighbor days through the classification models of random forest (RF) and KNN and then used the regression model of RF and KNN to predict the time of traffic congestion. Traditional machine learning is the simple linear regression model, which fails to capture the complex nonlinear spatial-temporal correlations.

Deep learning overcomes the limitations of the shallow learning of machine learning and can capture nonlinear spatial-temporal correlations well. Hopfield et al. [34] first proposed a time RNN model. With the in-depth study of the RNN, the model has been gradually applied to various fields. Yun et al. [35], used the Recurrent Neural Network (RNN) model for travel time prediction at expressway and urban intersections, and the experimental results showed that the model has good prediction performance. Since RNN has difficulty preserving long-term memory and has the problems of vanishing gradients and explosion gradients, researchers thus proposed to use Long Short-Term Memory (LSTM) [36], Gated Recurrent Unit (GRU) [37], and Bi-directional Long Short-Term Memory (BiLSTM) [38] for travel time prediction. However, it is difficult for a single neural network to capture both temporal and spatial correlations in a long time sequence. 

With the development of deep learning [39], combining the advantages of multiple single deep learning models into complex deep learning models has become a hot research topic in time series prediction [40]. Yao et al. [21] proposed a combined model for traffic flow prediction in which CNN and LSTM capture the spatial-temporal correlation of traffic flow and then use the periodically shifted attention mechanism to capture the periodicity and the flow gating mechanism to explicitly model dynamic spatial similarity. Liu et al. [41] also used a combined model for traffic flow prediction, which used Convolution LSTM to extract spatial-temporal correlations of traffic flow, and then used Bi-LSTM to capture periodicity. Guo et al. [42] combined CNN with LSTM to capture both temporal and spatial correlations in population flow prediction, and used temporal attention mechanisms to capture more important temporal information. Xu et al. [43] utilized feature embedding blocks to capture semantic information from multiple features. Then, based on the spatial attention mechanism and the temporal attention mechanism, captured the spatial and temporal dependencies in the multimodal traffic demand. In the above deep learning combined prediction model, they use the convolution idea to capture the spatial correlation, use the recurrent neural network to capture time correlation, and use the attention mechanism to capture important spatial-temporal information. 

At the same time, the combined model is also widely used in travel time prediction [44]. Li et al. [45] used CNN and LSTM to obtain spatial-temporal correlation, and then used the time attention mechanism to correct the drift error in travel time. Fang et al. [46] used a graph neural network (GCN) and a graph attention mechanism to obtain the spatial-temporal correlation of travel time and used CNN to obtain the spatial context information. Wang et al. [47] proposed a geo-based convolution, which converted the GPS series into a feature map, and then used LSTM to obtain the temporal correlation and a channel attention mechanism to capture the important information between different sub-paths. The above combined travel time models can capture both temporal correlations and spatial correlations, but they do not have an attention mechanism that considers both time and space. 

In summary, the difference of our proposed method compared with the literature is that we consider both spatial and temporal attention mechanism, and we also consider the travel time difference in vehicle types and the proximity of road network.

## 3. Methodology

### 3.1. Overview

In this section, we give an overview of the proposed model as shown in Figure 1. First, we preprocess the data to remove abnormal data and ensure the integrity of the data. On this basis, we converted the ETC data into vehicle trajectories based on the gantry topology data, so that we could obtain the travel time of vehicles in all sections. However, there is no traffic flow in a certain period of time in some sections, so there is no travel time in a certain time interval. Therefore, we supplemented the missing data by repairing the algorithm. After preparing the data, we analyzed and modeled multiple features, and process the vehicle separately according to the vehicle type. Finally, we used CNN to capture spatial correlation and BiLSTM to capture temporal correlation and then used the attention mechanism of spatial and temporal information to capture important information for model prediction. Therefore, we could obtain the predicted travel time of each vehicle.

### 3.2. Notations and Problem Formulation

In this section, we first fix some notations and define the travel time prediction problem. We follow previous studies [48] and define the set of time intervals as $I=\langle I_{1},\;I_{2},\ldots ,I_{t},I_{T}\rangle$. We further define the following:

**ETC data**. When the vehicle passes through the ETC gantry, the Road Side Unit (RSU) on the gantry will conduct an information transaction with the On Board Unit (OBU) of the vehicle. The RSU will record the vehicle ID, the gantry ID, the time of information transaction, the expressway entrance of the vehicle, and other information and then upload it to the ETC system. This uploaded information constitutes the ETC data Edata.

**Section**: The ETC gantry of the expressway is called Node, the area between two adjacent gantries forms a section which is referred to as $QD=\langle Dat,dis\rangle$, $Dat=\left\{Node_{1},Node_{2}\right\}$, where dis is the distance between two nodes. Node and QD are shown in Figure 2. The set of all sections (i.e., expressway network) can be express as $LW=\{QD_{1},\;QD_{2},\ldots ,QD_{n}\}$.

**Vehicle trajectory**: A set of ETC gantry Edata through which a vehicle passed while driving on the expressway, $Edata=\{RNode_{1},\;RNode_{2},\;RNode_{n}\}$, $RNode=\langle ID,\;Time,\;EnterID,\ldots .\rangle$. Edata are composed of multiple gantry transaction records, RNode. RNodes contain more than 100 data attributes; ID is the gantry Identity Document (ID); Time is the transaction time of the gantry; EnterID is the enter station ID. ETC data, Edata, can be converted into vehicle trajectory data, $Edata\to Traj=\left\{D_{0},D_{1},Di\ldots Dj,D_{E}\right\}$, $D_{i}=\left(N_{i},T_{i}\right)$, 0≤i≤E, $\forall i\leq j,\;T_{i}\leq T_{j}$. $D_{i}$ is the trajectory point, including node $N_{i}$ and time property $T_{i}$. $N_{i}$ is the label of the i-th node passed by the vehicle, and $T_{i}$ is the information interaction time when the vehicle passes through node $N_{i}$. $D_{0}$ is the start-point of the trajectory, and $D_{E}$ is the end-point of the trajectory.

**Vehicle type**: China’s license plates mainly include blue license plates, yellow license plates, green license plates, white license plates, and black license plates. In order to clarify the meaning of the vehicle type, the vehicle is divided into five categories according to the color of the license plate. They are Class A vehicle (blue license plate), Class B vehicle (yellow license plate), Class C vehicle (green license plate), Class D vehicle (white license plate), and Class E vehicle (black license plate). In addition, all vehicles together are called Class F vehicles.

**Travel time**: The time consumed by a vehicle passing a certain section $\langle Node1,\;Node2\rangle$ is called travel time Δt:(1)$$\Delta t=t_{node2}-t_{node1}$$

If m vehicles pass section $QD_{j}$ at time window i, the travel time $\Delta t_{i}^{j,all}$ of all vehicles at section $QD_{j}$ at time window i can be expressed as:(2)$$\Delta t_{i}^{j,all}=\left\{\Delta t_{i}^{j,1},\Delta t_{i}^{j,2},\ldots ,\Delta t_{i}^{j,m}\right\}$$

The average travel time Yij of the section QDj of m vehicles at time window i can be expressed as:(3)Yij=∑c=1mΔtijj,c/n
**Travel time prediction problem**: The travel time prediction problem aims to predict the travel time of t+1 time interval, given the data until time interval t. In addition to historical travel time data, we also include relevant context features, including spatial proximity features, spatial correlation features, time correlation features, and traffic situation correlation features. We define the context feature of section j at time point i as a vector eij∈ℝr, and r as the number of features. Therefore, travel time prediction can be expressed as:(4)Yi+1j=FYj−h,…jQD,…,Ej−h,…jQD

For j∈QD, $Y_{j-h,\ldots j}^{QD}$ is historical travel time, where j−h denotes the starting time. $F\left(.\right)$ is the prediction function. $E_{j-h,\ldots j}^{QD}$ are context features for all sections QD for time intervals from j−h to j. E can be expressed as:(5)$$E=\left[\begin{array}{c} E_{c}^{1},E_{c}^{2},\ldots ,E_{c}^{n}\\ E_{s}^{1},E_{s}^{2},\ldots ,E_{s}^{n}\\ \begin{array}{c} E_{t}^{1},E_{t}^{2},\ldots ,E_{t}^{n}\\ E_{z}^{1},E_{z}^{2},\ldots ,E_{z}^{n} \end{array} \end{array}\right]$$
where $E_{c}$ is the spatial proximity features, $E_{s}$ is the spatial correlation features, $E_{t}$ is the time correlation features, $E_{z}$ is the traffic situation correlation features, and $E^{n}$ denotes the context features from n-th time intervals.

### 3.3. Data Preprocessing

#### 3.3.1. Raw Data Cleaning

ETC gantries record transaction time, vehicle license plate, toll station, and other information when the vehicle passes through the ETC gantry. However, there are some special conditions (e.g., terrible weather, vehicle OBU anomaly, gantry RSU anomaly) that make the ETC system record abnormal data. Through research and analysis, the main abnormal data can be divided into data redundancy and data error, as shown in Table 1 and Table 2, where *, **, represents other characters that are not displayed.

**Data redundancy**: The transaction information of each vehicle passing through the ETC gantry should be unique. However, data collection, transmission, storage procedures may not work properly, resulting in multiple uploads of data. Therefore, these data need to be cleaned.

**Data error**: Data attributes differ from normal traffic data. There are three main cases: The first is that the data are not normally collected, which is replaced by special characters (e.g., Error 1). The second is that data are lost due to abnormalities in the system during transmission, and the system uses random characters to replace lost data (e.g., Error 2, Error 4). The third is that the data do not conform to normal traffic rules (e.g., Error 3), the time of the trade station being later than the time of the enter station. Therefore, these data need to be cleaned

#### 3.3.2. Vehicle Travel Time Construction

After preprocessing the ETC data, we construct the trajectory of each vehicle through the gantry sequence, and then calculate the section travel time of the vehicle combined with the gantry topology data, the details of the processing are shown in Algorithm 1. First, the trajectory of each vehicle is counted, and the vehicle trajectory is divided into multiple sections trajectory. Then, each section trajectory is matched with the gantry topology data to check whether the section belongs to the expressway gantry topology. If there is, the travel time of the vehicle passing through the section is directly calculated; if not, we need to search for the shortest path of the section, through the shortest path of the section sequence, to add the missing gantry record. After the data are added, we can calculate the travel time of all sections. The algorithm is as follows.
**Algorithm 1** Travel time window construction algorithm.**Input:** ETC data Edata; Expressway road network topology data LW;**Output:** Vehicle travel time data Dt;1: Edata = $\left\{RNode_{1},RNode_{2},\ldots ,RNode_{n}\right\},$ $LW=\left\{QD_{1},QD_{2},\ldots ,QD_{n}\right\}$, $QD=\langle Dat,Dis\rangle$;2: **for** i = 0 to i = n−1 **do**3: $\Delta t_{i}$ = $Time_{i+1}-Time_{i}$//Calculating the time difference of adjacent nodes;4: $Dat=\langle RNode_{i},Time_{i},RNode_{i+1},Time_{i+1}\rangle$//save the information of adjacent nodes;5: Dt = $\left(Dat,\Delta t_{i}\right)$,//save the vehicle passage information;6: **end for**7: **if** $RNode_{i}$ and $RNode_{i+1}$ in LW//if adjacent nodes are in topological data;8: Dt = $\left(Dat,\Delta t_{i}\right)$//the vehicle passage time remains unchanged;9: **else**10: Dis = {}11: $\left\{N_{1},N_{2},\ldots ,N_{m}\right\}$ ← shortest path(LW,N)//search for the shortest path, which $N=\left\{RNode_{i},\;RNode_{i+1}\right\}$;12: $Dis\leftarrow \left\{N_{1},N_{2},\ldots ,N_{m}\right\}$//the shortest distance is converted into distance;13: $v_{i}=Dis/N_{i.Time}$//calculate the speed of the front and back gantry;14: $\Delta t_{i}=N_{i+1.Time}-N_{i.Time}$15: $Dat=\langle N_{i},N_{i.Time},N_{i+1},N_{i+1.Time},\Delta t\rangle$16: Dt = $\left(Dat,\Delta t_{i}\right)$,//save the vehicle passage information;17: Return Dt


#### 3.3.3. Repair of Missing Data of Time Interval

In real traffic conditions, some sections with small traffic flow will have no traffic flow at a certain time interval, so there is no travel time feature in these time intervals. To solve this problem, some researchers will add an ideal value to replace missed data. Since the ideal value cannot truly reflect the traffic situation, it may cause some problems when predicting the travel time. Therefore, we filled the missing value according to the historical travel time correlation of the road network, the details of the processing are shown in Algorithm 2.
**Algorithm 2** The addition algorithm of missing data in time window.**Input**: $data=\left\{x_{1},x_{2},x_{3}\ldots ,x_{n}\right\}$//The sequences with missing values;**Output**: $data^{*}=\left\{x_{1},x_{2},x_{3},x_{4}\ldots ,x_{n}\right\}$//The complete sequence;**1: for** i ← 0 to n **do**2:  **if** i==03:   **if** $data\left[i\right]$ is nan and $\;data\left[i+1\right]$, data $data\left[i+2\right]$ is not nan4:    $data\left[i\right]\leftarrow \left(\mathrm{data}\left[i+1]+\mathrm{data}[i+2\right]\right)/2$;5:   **end if**6:  **end if**7:  **if** i==18:   **if** $data\left[i\right]$ is nan and $data\left[i-1\right]$, $data\left[i+1\right]$ is not nan9:    $data\left[i\right]\leftarrow \left(\mathrm{data}\left[i-1]+\mathrm{data}[i+1\right]\right)/2$;10:   **end if**11: **end if**12:  **if** i>=2 and $i<=\left(n-3\right)$13:   **if** $data\left[i\right]$ is nan and $data\left[i-1\right]$, $data\left[i+1\right]$ is not nan14:    $data\left[i\right]\leftarrow \left(\mathrm{data}\left[i-1]+\mathrm{data}[i+1\right]\right)/2$;15:   **end if**16:   **if** $data\left[i\right]$, $data\left[i+1\right]$ is nan and $data\left[i-1\right]$, $data\left[i-2\right]$ is not nan:17:     $data\left[i\right]\leftarrow \left(\mathrm{data}\left[i-1]+\mathrm{data}[i-2\right]\right)/2$;18:   **end if**19:   **if** $data\left[i\right]$, $data\left[i-1\right]$ is nan and $data\left[i+1\right]$, $data\left[i+2\right]$ is not nan20:    $data\left[i\right]\leftarrow \left(\mathrm{data}\left[i+1]+\mathrm{data}[i+2\right]\right)/2$;21:   **end if**22:  **end if**23:  **if** i==n−224:   **if** $data\left[i\right]$ is nan and $data\left[i-1\right]$, $data\left[i+1\right]$ is not nan25:    $data\left[i\right]\leftarrow \left(\mathrm{data}\left[i-1]+\mathrm{data}[i+1\right]\right)/2$;26:   **end if**27:   **if** $data\left[i\right]$, $data\left[i+1\right]$ is nan and $data\left[i-1\right]$, $data\left[i-2\right]$ is not nan28:     $data\left[i\right]\leftarrow \left(\mathrm{data}\left[i-1]+\mathrm{data}[i-2\right]\right)/2$;29:   **end if**30:   **end if**31:  **if**
i==n−132:   **if** $data\left[i\right]$ is nan and $data\left[i-1\right]$, $data\left[i-2\right]$ is not nan33:    $data\left[i\right]\leftarrow \left(\mathrm{data}\left[i-1]+\mathrm{data}[i-2\right]\right)/2$;34:   **end if**35:  **end if**36: **end for**37: $data^{*}\;$← data38: **return** $data^{*}$


The algorithm mainly repairs the missing values of the first part, the middle part, and the end part of the data and then supplements them according to the correlation of the travel time before and after the section. The repair effect is shown in Figure 3. Figure 3 shows that the repair value will be dynamically supplemented according to the correlation before and after, and the added value can almost reflect the real traffic situation.

### 3.4. Travel Time Analysis and Modeling

#### 3.4.1. Differentiation Analysis of Vehicles

There are many different types of vehicles on expressways in China, and the travel time of different types of vehicles may be different. Therefore, we analyze the travel time of two sections with long mileage and short mileage, and the distribution of travel time is shown in Figure 4. It can be drawn from the figure that the travel time of different types of vehicles is different, and the travel time of Class B vehicles is the largest difference with other types of vehicles, and its travel time value is much higher than the average travel time of all vehicles (Class F vehicle). The overall average travel time is almost in the middle between the Class B vehicles and other types of vehicle, which is smaller than that of the Class B vehicles and larger than that of other types of vehicles. Therefore, it can be concluded that there are differences in travel time among different types of vehicles. It is irrational to predict travel time with the overall travel time, and it is necessary to analyze the travel time of each vehicle separately.

To effectively obtain the difference between each type of vehicle, we further analyze the above Section 1 and Section 2. Figure 5 shows the travel time comparison of vehicles in two sections, and Figure 6 shows the average absolute error results of each type of vehicle. It can be drawn from Figure 5 that in addition to the vehicles with Class B vehicle, the travel time values of all types of vehicles have little difference, and the travel time of a Class B vehicle is significantly higher than that of other vehicle types. Figure 6 shows that the absolute error of travel time between Class B vehicle and other vehicles is relatively large, and the average absolute error between other types of vehicles is relatively small. Therefore, Class B vehicles and other types of vehicles need to be processed separately. The travel time of vehicles is mainly between Class II vehicles (Class B vehicle, just that, big vehicles with yellow license plates) and Class I vehicle (Class A vehicle, Class C vehicle, Class D vehicle, Class E vehicle, just that, small vehicles with other color license plates). we mainly construct two travel time prediction models of Class I and Class II vehicles to realize the travel time prediction of expressways.

#### 3.4.2. Context Features Modeling

In this section, we will introduce each context feature in detail.

(1)Spatial proximity features

Near things are more related than distant things [49]. There is a strong continuity between the sections of the expressway, so the flow and speed between the sections are correlated in the road network [9]. However, each section of the road network has different distances, and the travel time of the upstream and downstream sections will also be different. It is difficult to directly capture the correlation between adjacent sections. Therefore, we consider converting the speed proximity of adjacent sections to the travel time proximity. We denote the section speed as $V_{QD\left(i\right)}$, velocity has the correlation of adjacent sections, $V_{j-r}\propto \ldots \propto V_{j}\propto \ldots \propto V_{j+r}$, and the section distance $Dis_{QD\left(i\right)}$ is fixed. Therefore, we construct the travel time proximity $E_{c}$ based on $Dis_{QD\left(i\right)}$ and $V_{QD\left(i\right)}.\;E_{c}$ can be represented as:(6)$$E_{c}=\frac{Dis_{j}}{V_{j\pm r}}$$
where j is the gantry number, and r is the number of adjacent gantries.

The r value is the key to spatial proximity. To obtain the optimal range of r, we use Pearson’s correlation coefficient to analyze the influence factors of travel time data $T_{j\pm r}$ on $T_{j}$. The calculation formula is:(7)$$\rho =\frac{\sum_{j=1}^{n}\left(T_{j}-\bar{T_{j}}\right)\left(T_{j\pm r}-\bar{T_{j\pm r}}\right)}{\sqrt{\sum_{i=1}^{n}{\left(T_{j}-\bar{T_{j}}\right)}^{2}\sum_{i=1}^{n}{\left(T_{j\pm r}-\bar{T_{j\pm r}}\right)}^{2}}}$$
where n represents the number of traffic samples, $T_{j}$ is the travel time of section $QD_{j}$, and $T_{j\pm r}$ is the travel time of r sections before and after section j.

We selected three sections for spatial proximity analysis. As shown in Table 3, it can be concluded that with the increase in section distance, the correlation of proximity decreases gradually. There is a strong correlation between the two sections.

(2)Spatial correlation features

In the road network, different sections have similar traffic speeds and zone mileages; that is, they have similar travel times, so the travel time is a spatial correlation. A large amount of traffic situation prediction research (e.g., traffic flow, traffic speed, travel time) has consider spatial correlation. Therefore, in this study, spatial correlation is also considered in the travel time prediction. The spatial correlation $E_{s}$ can be expressed as the similar travel time of section $QD_{c}$ and section $QD_{j}$ and section $QD_{z}$ in the road network, $T_{c}\propto T_{j}\propto T_{z}$.

(3)Temporal correlation features

There is temporal correlation in travel time. The temporal correlation can be expressed as $E_{t}=\left\{E_{w},E_{d},E_{tc}\right\}$. $E_{d}$ is the daily periodicity,$\;E_{d}$ is the weekly periodicity, and $E_{tc}$ is the time closeness. The travel time of a certain day in each week will have a weekly periodicity (e.g., Friday is the day before the weekend holiday in China, a large number of people will go back to their hometown, Sunday is the last day of the holiday, and a large number of people will return to the city to work), a certain hour in each day also has periodicity (e.g., 7–9 p.m. is the peak time of traveling to work, 5–7 p.m. is the peak time of getting off work). At the same time, there is also a close correlation between the previous time intervals and the latter time interval.

(4)Traffic situation features

Traffic situations consist mainly of traffic flow, travel time, and traffic speed, which interact with each other (e.g., traffic flow increases to a certain level, traffic speed becomes smaller, and travel time becomes larger). Therefore, traffic situation features are expressed as $E_{z}=\left\{E_{v},E_{q}\right\}$, where $E_{v}$ is traffic speed, and $E_{q}$ is traffic flow.

### 3.5. Deep Learning Prediction Model

This section provides details about the deep learning framework. which aims to predict the expressway section travel time $Y\left(t\right)$ at the next moment by using nearby historical travel time data $R_{closs}$, spatial proximity data $R_{c}$, traffic situation data $R_{z}$, and periodic data $R_{d},\;R_{w}$. Figure 7 shows the architecture of the deep learning model.

First, we input spatial proximity data $R_{c}$ and traffic situation data $R_{z}$ into the CNN module to capture the correlation of feature space and road network proximity and then use the level attention of CBAM attention mechanism to weight important features to improve the prediction performance of the model.

Second, we also use CNN to obtain the spatial correlation of the road network, and then use the spatial attention of the CBAM attention mechanism to obtain important spatial information.

Finally, we input the periodic data $R_{d}$ and $R_{w}$ into the BiLSM module to capture the proximity of time and the periodicity of time. Then, the time attention mechanism is used to dynamically adjust the weight of each time interval on the prediction results.

#### 3.5.1. CNN-ATTENTION

In the road network, some sections may have similar travel time distribution, and adjacent regions also have some correlation. Therefore, we use the CNN module to capture the correlation and proximity in the road network space. At the same time, the traffic flow and traffic speed of the section are also potentially correlated with travel time. Therefore, we use the CNN model to capture the correlation between traffic flow and traffic speed and travel time. The convolution neural network inputs the extracted original input $Y_{i,t}$ as $Y_{i,t}^{k}$ into the convolution layer k, and uses two-dimensional convolution to capture the spatial feature of the travel time of the section. The convolution formula is shown in follows:(8)$$Y_{i,t}^{k}=ReLU\left(W^{k}*Y_{i,t}^{k-1}+b^{k}\right)$$
where *K* is the number of convolution layers, i is the section, t is the time window, ReLU is the activation function, $W^{k}$ is the weight coefficient, and $b^{k}$ is the constant.

To further capture important information in the spatial dimension, we use the CBAM attention mechanism [50] to fully understand the detail changes in features and further enhance local spatial feature representation. In level attention, CBAM can strengthen the travel time distribution on the feature map, so that the model can obtain the most important features and give greater weight to travel time prediction.

Precisely, as shown in Figure 8, in the level feature attention calculation, we first use CBAM for global max-pooling and average-pooling of the inputs by level to obtain Maxpool level attention vector and AvgPool level attention vector. Then, we input these two vectors into a single-layer perceptron with shared weights to obtain two new hierarchical attention vectors. We combine these two vectors by the sum of elements and multiply them with the original feature map to obtain a new feature map, which can be expressed as:(9)$$M_{c}=\sigma \left(MLP\left(Maxpool\left(Y_{i,t}^{k}\right)\right)+MLP\left(AvgPool\left(Y_{i,t}^{k}\right)\right)\right)$$
(10)$$F^{c}=M_{c}\left(Y_{i,t}^{k}\right)⨂Y_{i,t}^{k}$$
where ⨂ is the multiply operation, and σ represents the sigmoid function

At the same time, different sections have different levels of influence on the travel time prediction of the predicted sections. CBAM can capture important sections in the spatial dimension and give greater prediction weights, the spatial attention mechanism is shown in Figure 9. In the spatial attention calculation of the road network, we first perform average-pooling and max-pooling operations along the level axis to obtain two spatial attention maps and connect them. Then, a standard convolution layer is used to connect them, and convolution operations are used to generate a spatial attention weight matrix. Finally, the weighted feature matrix is obtained by multiplying the matrix with the input feature mapping, which is expressed as:(11)$$M_{s}=\sigma \left(Conv\left(\left[Maxpool\left(Y_{i,t}^{k}\right);AvgPool\left(Y_{i,t}^{k}\right)\right]\right)\right)$$
(12)$$F^{s}=M_{s}\left(Y_{i,t}^{k}\right)⨂Y_{i,t}^{k}\;$$
where ⨂ is the multiply operation, and σ represents the sigmoid function.

#### 3.5.2. BiLSTM-ATTENTION

There are some rules in daily life: travel time usually shows obvious periodicity and trends. Therefore, this section will focus on periodic and trends in travel time series. We use BiLSTM to capture the proximity correlation and periodic correlation for historical travel time. The input of long time series travel time data into BiLSTM will make it difficult to learn a reasonable vector representation, thus affecting the prediction performance of the model. Therefore, we use the time attention mechanism to capture important information in time to improve the prediction performance of the model.

The LSTM network is a kind of time-recurrent neural network with memory characteristics. It is a variant of RNN, which can effectively overcome the long-term dependence and gradient vanishing of RNN. To effectively obtain the context correlation before and after, BiLSTM is proposed. BiLSTM is composed of forward LSTM and backward LSTM, and the two LTSM models work on the same principle and have the same internal structure. The BiLSTM model is shown in Figure 10.

Each moment in the Bi-LSTM model is jointly determined by the state of LSTM in two directions, and the Bi-LSTM calculation formula is:(13)$${\vec{h}}_{t}=LSTM\left(x_{t},{\vec{h}}_{t-1}\right)$$
(14)$${\overset{\leftarrow }{h}}_{t}=LSTM\left(x_{t},{\overset{\leftarrow }{h}}_{t-1}\right)$$
(15)$$h_{t}=w_{t}{\vec{h}}_{t}+v_{t}{\overset{\leftarrow }{h}}_{t}+b_{t}$$
where $w_{t}$ is the weight coefficient of each output in the forward LSTM model, and then the weight matrix is constructed; $v_{t}$ is the weight matrix constructed by the weight coefficient of each output in the backward LSTM model; and $b_{t}$ is the bias at time t.

After capturing the correlation between the historical travel time and the last travel time, we use BiLSTM to predict the results of the t time window using the time series of the previous t−1 time window. The prediction formula is expressed as:(16)$$h_{i,t}=BiLSTM\left(Y_{i,t},h_{i,t-1}\right)$$
where $h_{i,t}$ denotes the prediction result of section i in time window t, and $Y_{i,t}$ is the input value.

For the daily periodicity and weekly periodicity of travel time, we use BiLSTM to obtain the correlation of n days. The formula can be expressed as:(17)$$h_{i,d}^{n}=BiLSTM\left(Y_{i,d}^{n},Y_{i,w-1}^{n},h_{i,d-1}^{n}\right)$$
where $h_{i,d}^{n}$ is the travel time prediction results of d hours in section i.

At the same time, the contribution of the previous n day historical travel time to the prediction is not equal. For example, the effect of yesterday on prediction performance is more significant than that of other days at the same time. In addition, there is travel restrictions in some areas (e.g., in some areas or time, vehicle travel is restricted), people travel more similar on alternate days. In the same way, there are similar travel patterns in the daily periodicity. To effectively capture the important information of daily periodicity and weekly periodicity, we use the time attention mechanism to assign different weights to daily (weekly) travel time from potential daily periodicity and potential weekly periodicity. the equation of attention contribution weight can be shown as:(18)$$a_{i,d}^{n}=\frac{\exp\left(s\right)}{\sum_{d\in D}\exp\left(s\right)}$$
where $a_{i,t}^{n}$ is the importance of section i at time window t on day p, D is the number of time intervals for input, and s is the contribution scoring function, which can be expressed as:(19)$$s=v*\tanh\left(h_{i,d}^{n}w_{H}+h_{i,t}w_{X}+b\right)$$
where $w_{H},\;w_{X}$, b, v are learned parameters.

Finally, the output value of BiLSTM is used as the input of attention mechanism to predict the travel time, and the calculation formula is shown:(20)$$h_{i,t}^{n}=\sum_{d\in D}a_{i,d}^{n}*h_{i,d}^{n}$$

## 4. Results

### 4.1. Experimental Settings and Data Description

The experimental conditions of this experiment are the Window 10 system, a Lenovo computer equipped with an Intel kernel, 2.6 GHz processor, and 16 GB memory; all experiments use Python 3.7 version, and the software architecture is developed based on Keras deep learning library tool.

The ETC data used in this experiment are from Fujian Provincial Expressway Information Technology Co., Ltd. (Fuzhou, China), which were collected by ETC gantry system from 3 May 2021 to 3 June 2021. The transaction data contain 103 attributes, including license plate number, enter time, enter station, gantry transaction time, and gantry latitude and longitude, the details is shown in Table 4, where *,**,*** represents other characters that are not displayed. In order to verify the effectiveness of the proposal, the Fuzhou-Xiamen Expressway, with the largest traffic flow in Fujian Province, is selected as the source of experimental data. It includes the four cities of Fuzhou, Putian, Quanzhou, and Xiamen. We used 70% of the data as training data and 30% as test data. The position of the gantry is shown in Figure 11.

### 4.2. Evaluation Metric

We evaluate the predictive performance of our proposed method and existing methods with two widely-applied metrics. They are Mean Absolute Error (MAE) and Root Mean Square Error (RMSE), respectively. The calculation formula is as follows:(21)$$\mathrm{MAE}=\frac{1}{q*p}\sqrt{\overset{m}{\underset{i=1}{\sum }}\left|Y_{i}^{\prime }-Y_{i}\right|}$$
(22)$$\mathrm{RMSE}=\sqrt{\frac{1}{q*p}\overset{m}{\underset{i=1}{\sum }}{\left(Y_{i}\prime -Y_{i}\right)}^{2}}$$
where q is the number of sections in the experiment, p is the number of all time windows, $Y^{\prime }$ is the predicted value, and $Y_{i}$ is the actual value.

### 4.3. Analysis of Sequence Length

The Sequence length has great influence on travel time prediction. To further analyze the performance of the sequence length during prediction, the sequence lengths were tested from 1 to 9, and Figure 12a shows the trend of MAE as the sequence length increases. Figure 12b shows the trend of RMSE. From Figure 12, we can know that the sequence length is relatively better when it is 4 and 5, where 4 is the best. Therefore, we use four time intervals to predict the next time interval.

### 4.4. Analysis of Classification Based on Vehicle Type

In order to verify the essentiality of considering the type of vehicle, we use the MVTTP model to predict the travel time of Class I vehicles (i.e., big vehicles) and Class II vehicles (i.e., small vehicles). The number of convolution kernels is 64, the CNN and BiLSTM activation function are both ReLU, the Optimizer is adam, the cell of BiLSTM is 50, the batch size is 30, and the training epoch is 50. We set the time window to 20 min, taking the previous four time windows as input windows to predicting the next 20 min.

We test the performance of Class I vehicle and Class II vehicle after classification, the results are shown in Table 5. Table 5 shows that there is a big difference between the travel time prediction performance without considering the vehicle type and considering the vehicle type. The prediction performance of the MVTTP model in the two vehicle types is much better than that of the vehicle type without considering the vehicle.

Figure 13 is the travel time visualization of four sections, which shows that there is a big difference between the predicted travel time without considering and the real travel time of the vehicle in most of the time. For the section with a travel time of about 100 s, the prediction error is about 10 s, while for the section with a longer travel time, the error is about 100 s, and the error is about 10%. The predicted travel time without considering the type of vehicle is much higher than the predicted travel time of Class II vehicles. This is because Class II vehicles will be faster than other types of vehicles, and travel time will be shorter. Only in the peak period of travel is the speed of the vehicle reduced, and the travel time of all vehicles is similar. The difference between the predicted value without considering the type of vehicle and the real travel time of the Class II vehicles will be reduced. For the travel time prediction of Class I vehicles, the travel time is far less than the predicted value without considering the type of vehicle. Compared with Class II vehicles, the real travel times of Class I vehicles have a bigger difference from the predicted value without considering the vehicle type. The error of all sections is greater than 10% most of the time, and only in the peak period will the gap narrow.

### 4.5. Analysis of Spatial Proximity

In this section, we test the predictive performance of spatial proximity features on the model. The test results are shown in Table 6. Table 6 shows that the prediction performances of both Class I vehicles and Class II vehicles are improved by the spatial proximity. The prediction performance of the MVTTP model can be further improved by considering the spatial proximity.

### 4.6. Analysis of Spatial-Temporal Attention Mechanism

We test the performance of temporal attention mechanisms and spatial attention mechanisms, and the test results are shown in the Table 7. Table 7 shows that the prediction performance of the MVTTP is further improved after considering the spatial attention mechanism or temporal attention mechanism. When both attention mechanisms are considered, the model has better prediction performance.

### 4.7. Comparative Analysis of Prediction Models

To analyze the prediction performance of our proposed model (MVPPT), we use the classified vehicle data to compare the following methods, it includes traditional time series prediction methods, machine learning algorithms, and the current best deep learning processing model.

**HA:** Historical Average, the traditional time-series prediction methods, which predicts the travel time using average values of previous travel time values at the location given in the same relative time interval.

**KNN**: K-Nearest Neighbor, which is one of the most classical classification and regression methods in data mining.

**SVR**: Support vector regression model applies the support vector machine (SVM) similarity method for regression analysis.

**AdaBoost**: Adaptive Boosting. AdaBoost is a robust boosting tree-based method that is widely used in data mining applications. 

**LSTM**: Long Short-Term Memory, a kind of time-recurrent neural network, which is good at processing time series data.

**CNN**: Convolutional Neural Network, which is widely used to capture the spatial correlation of time series for time series prediction.

**BiLSTM** [51]: Bi-directional Long Short-Term Memory, which is composed of forward LSTM and backward LSTM.

**TGCN** [52]: Time Domain Graph Convolutional Network, which is a well-known traffic forecasting method.

**STDN** [21]: Spatial-Temporal Dynamic Network, a method to jointly model both spatial and temporal dependencies by integrating CNN and LSTM.

The results of the travel time prediction test of the model in Table 8 show that the traditional travel time prediction algorithm (HA) has the worst prediction performance, because it predicts values only according to historical records without considering the context feature. Compared with the traditional time series prediction method, the machine learning method considering multidimensional feature of the travel time and has better prediction performance, among which SVR has the best performance. However, machine learning cannot capture nonlinear spatial and temporal correlation. Therefore, the neural network (e.g., LSTM, BiLSTM), which can capture spatial-temporal information, has better prediction performance, and only CNN has slightly worse prediction performance than SVR. In addition, we also use TGCN for performance testing. Due to the different distances of each road, the travel time is also different. TGCN is difficult to captures sufficient correlation between roads, and only time correlation can be captured. Therefore, TGCN did not have a very good prediction performance. In contrast, STDN handles spatial and temporal information via local CNN and LSTM, and using a periodically shifted attention mechanism to learn the long-term periodic dependency, which have a better prediction performance. Our proposed model (MVPPT) uses CNN and LSTM to capture the spatial-temporal context information, using the attention mechanism to capture important information. In addition, MVPPT also consider the context feature of spatial proximity, so it has the best prediction performance.

## 5. Conclusions

For expressway travel time prediction, we analyze the travel times of different types of vehicles and propose a novel model of expressway section travel time prediction. From the analysis and experimental results, we can find:(1)There are big differences in travel time among all types of vehicles. The travel time of big vehicles with yellow license plates is much longer than others types of vehicles. The main difference in travel time can be divided into two categories: big vehicles with yellow license plates and small vehicles with the rest of the plate colors.(2)The predicted travel time without considering vehicle type is higher than the real travel time of small vehicles and smaller than the real travel time of big vehicles. The error of travel time prediction without considering the type of vehicle is about 10%. After considering the type of vehicle, the prediction performance of the model has been significantly improved, and the predicted values of the model are close to the real travel time values of the vehicle.(3)The expressway network has close proximity, and the travel time prediction model can further improve the prediction performance after using the road network proximity. At the same time, the temporal attention mechanism and spatial attention mechanism can capture more important information, which can further improve the prediction performance of the model, and the model combining the two attention mechanisms has the best prediction performance.(4)This proposal can accurately predict the travel time of each section, which is of great significance for the fine management of the expressway and the development of smart expressways.

This proposal can accurately predict the travel time of each section, which is of great significance to the fine management of highways and the development of smart highways.

In fact, there are still many undiscovered rules about expressway travel times. In the future, we can further improve the performance of the model by capturing the monthly periodicity and holiday periodicity in larger datasets.

## 6. Discussion

The prediction of travel time of a section can promote the fine management of expressways and provide more accurate travel times to the people using them. Therefore, we model the travel time prediction based on ETC data, considering the difference of travel time of different types of vehicle and the proximity of expressway network. At the same time, we also consider the spatial-temporal attention mechanism in the deep learning framework, which constitutes a multi-view travel time prediction model. The experimental results verify the effectiveness of the model. However, the proposal also has some local specificities. First, the experimental data can only be a complete data set that includes all the type of vehicles so that the types of vehicles can be classified, which is difficult because only the relevant transportation departments can collect these data. Second, the road network proximity proposed in this work may only be applicable to the travel time prediction of expressways. The urban traffic network is complex, and the correlation between adjacent road sections is small. Therefore, road network proximity cannot provide a large contribution in travel time predictions in cities.

## Figures and Tables

**Figure 1 entropy-24-01050-f001:**
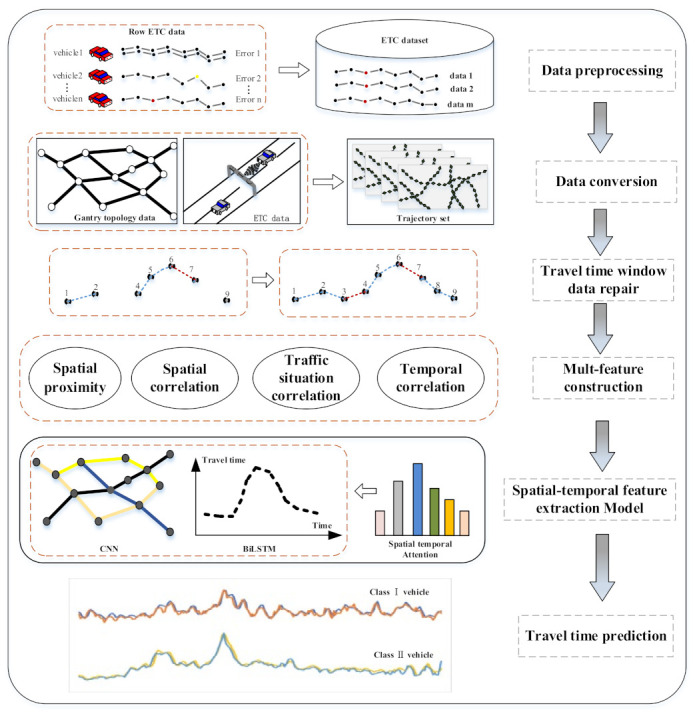
Overall framework.

**Figure 2 entropy-24-01050-f002:**
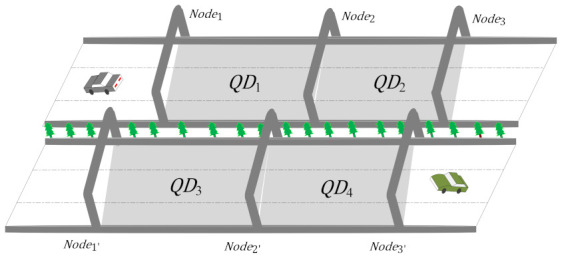
Schematic of the sections.

**Figure 3 entropy-24-01050-f003:**
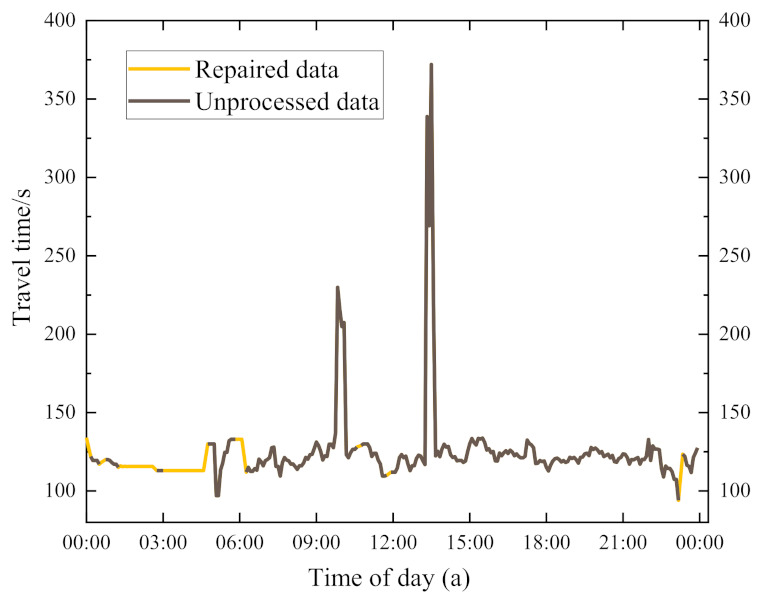
Repair effect of algorithm.

**Figure 4 entropy-24-01050-f004:**
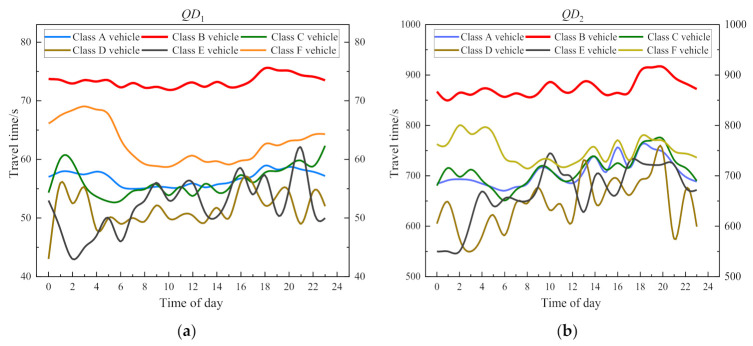
Travel times visualization of all types of vehicles: (**a**) is a visualization of section 1; (**b**) is a visualization of section 2.

**Figure 5 entropy-24-01050-f005:**
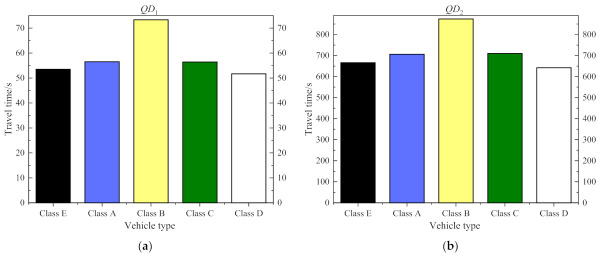
Travel time statistics of different types of vehicles: (**a**) is the statistics for section 1; (**b**) is the statistics for section 2.

**Figure 6 entropy-24-01050-f006:**
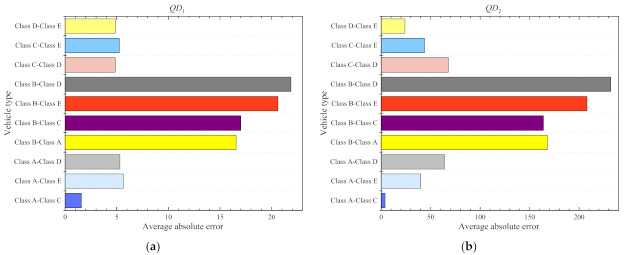
Average absolute error of travel time between different types of vehicles: (**a**) is the statistics for section 1; (**b**) is the statistics for section 2.

**Figure 7 entropy-24-01050-f007:**
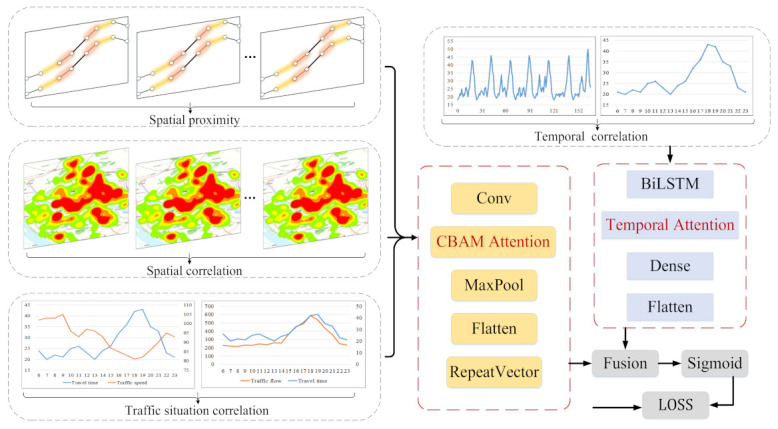
Deep Learning Prediction Framework.

**Figure 8 entropy-24-01050-f008:**
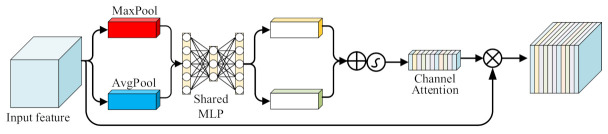
The level attention.

**Figure 9 entropy-24-01050-f009:**
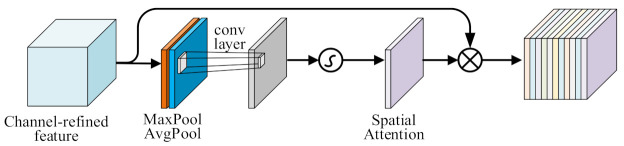
The spatial attention.

**Figure 10 entropy-24-01050-f010:**
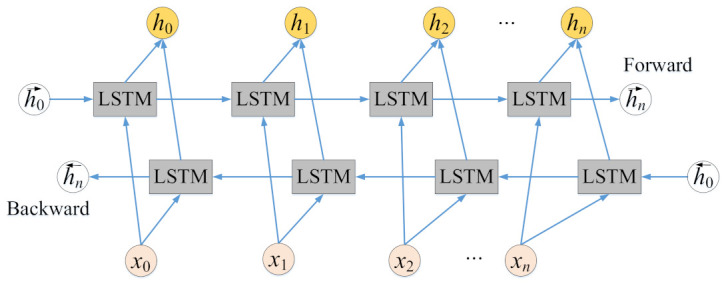
The BiLSTM framework.

**Figure 11 entropy-24-01050-f011:**
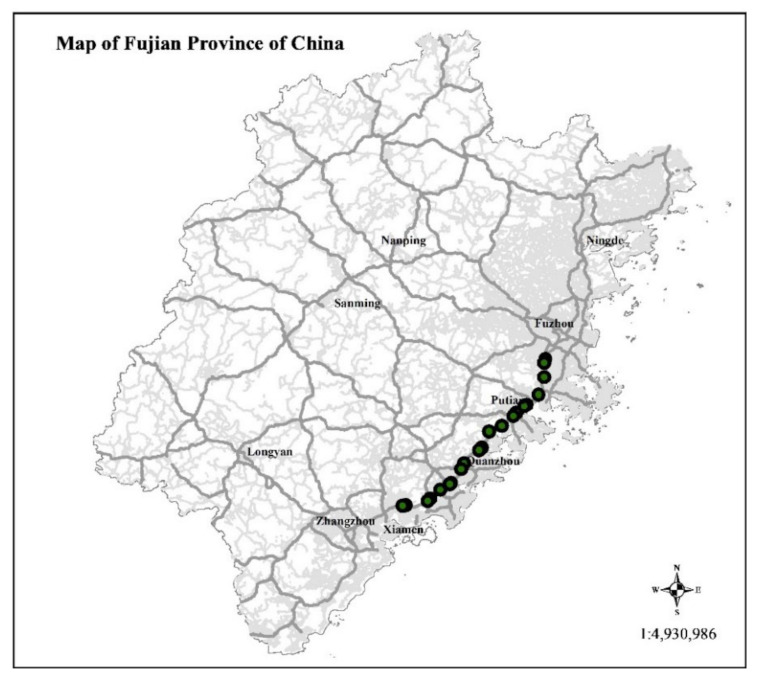
Distribution of gantries in Fuzhou-Xiamen Expressway.

**Figure 12 entropy-24-01050-f012:**
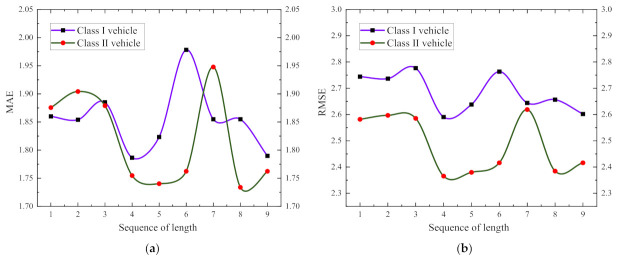
Analysis for between different sequence lengths: (**a**) is the MAE; (**b**) is the RMSE.

**Figure 13 entropy-24-01050-f013:**
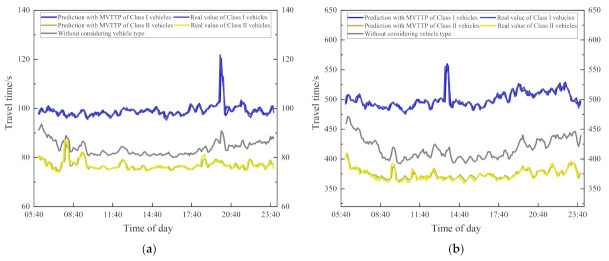
Visualization of travel time prediction, (**a**) is a visualization of section 1; (**b**) is a visualization of section 2; (**c**) is a visualization of section 3; (**d**) is a visualization of section 4.

**Table 1 entropy-24-01050-t001:** Examples of data redundancy.

Tradeid	Obuid	Tradetime	Flagid	Carplate	…
G001639 **	6A59 **	27 May 2021 6:21:38	3402 *	Blue MinA12	…
G001639 **	6A59 **	27 May 2021 6:21:38	3402 *	Blue Min A12	…
G001639 **	6A59 **	27 May 2021 6:21:38	3402 *	Blue Min A12	…
G001639 **	6A59 **	27 May 2021 6:21:38	3402 *	Blue Min A12	…

**Table 2 entropy-24-01050-t002:** Examples of data error.

Class	Obuid	Entime	Flagid	Ttradetime	…
Error 1	62F3 **	**000000**	3502 *	20 May 2021 11:21:38	…
Error 2	6873 **	22 May 2021 7:31:54	**a6p823**	22 May 2021 13:11:50	…
Error 3	628A **	25 May 2021 8:21:38	350A *	25 May 2021 **0:56:32**	…
Error 4	**236d45**	29 May 2021 9:29:11	3502 *	29 May 2021 15:23:11	…

**Table 3 entropy-24-01050-t003:** Pearson’s correlation analysis of adjacent sections.

**Adjacent sections**	$T_{j-1}$	$T_{j-2}$	$T_{j-3}$	$T_{j-4}$
ρ	0.63	0.59	0.36	0.32
**Adjacent sections**	$T_{j+1}$	$T_{j+2}$	$T_{j+3}$	$T_{j+4}$
ρ	0.60	0.51	0.38	0.263

**Table 4 entropy-24-01050-t004:** ETC data attribute.

Attribute Name	Examples	Attribute Name	Examples
Trade ID	452 *** 56	OBU Plate	Blue Fujian A1 ** 45
Trade time	6 September 2020 21:29:26	Vehicle Class	1
Flag ID	33 ** 21	Enter Time	6 September 2020 20:23:51
Flag Type	0	Enter Station	16 * 7
Flag Index	1	OBU ID	11C *** B6
LAT	118.39 **	LNG	24.66 ***

**Table 5 entropy-24-01050-t005:** Performance of the considering vehicle type.

Model	Class II Vehicles		Class I Vehicles	
	MAE	RMSE	MAE	RMSE
Unconsidering vehicle type	36.3128	57.9982	59.3436	84.8430
**MVPPT**	**8.50313**	**19.1132**	**11.5529**	**18.6298**

**Table 6 entropy-24-01050-t006:** Performance of the considering spatial proximity.

Model	Class II Vehicle		Class I Vehicle	
	MAE	RMSE	MAE	RMSE
MVPPT without spatial closeness	9.0033	21.7577	11.8418	19.6733
**MVTTP**	**8.50313**	**19.1132**	**11.5529**	**18.6298**

**Table 7 entropy-24-01050-t007:** Test results of spatial-temporal attention mechanisms.

Model	Class II Vehicles		Class I Vehicles	
	MAE	RMSE	MAE	RMSE
MVTTP without any Attention	8.9798	19.9776	11.8078	19.4728
MVTTP without spatial Attention	8.890931878	19.80547952	11.6711	19.4977
MVTTP without temporal Attention	8.852875979	19.86132629	11.6877	19.4169
**MVTTP**	**8.50313**	**19.1132**	**11.5529**	**18.6298**

**Table 8 entropy-24-01050-t008:** Performance of prediction models.

Model	Class II Vehicles		Class I Vehicles	
	MAE	RMSE	MAE	RMSE
HA	19.7064	37.3719	23.1210	33.7131
KNN	15.9966	31.5796	18.2482	28.7021
SVR	11.8366	23.1494	14.9408	20.9557
AdaBoost	12.464	28.9111	13.7415	21.3218
CNN	12.6426	26.7706	15.5382	24.5275
LSTM	9.7911	20.8325	12.0161	19.4116
BiLSTM	9.5706	22.7144	11.8629	19.2269
TGCN	14.7399	30.7650	16.4463	31.8081
STDN	9.3075	20.5715	11.9132	19.3659
**MVPPT**	**8.50313**	**19.1132**	**11.5529**	**18.6298**

## Data Availability

Restrictions apply to the availability of these data. Data were obtained from Fujian Expressway Information Technology Co., Ltd. (Fuzhou, China), and are available from the authors with the permission of Fujian Expressway Information Technology Co., Ltd.

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
