# Peer review of "Multi-View Travel Time Prediction Based on Electronic Toll Collection Data"

_entropy, 2022, doi:10.3390/e24081050_

Round 1

Reviewer 1 Report

The present paper focusing on Travel Time Prediction Based on Electronic Toll Collection Data is well put together, but has quite significant potential for improvement. For example, in the "Introduction" chapter, I lack references to relevant sources that would support the theses mentioned here. The "Related Work" chapter, which substitutes for the literature search, is very short and I miss sources dealing with mobility and its quantification using different concepts (e.g. the effect of rhythmicity of places). The authors almost exclusively cite other Chinese authors and completely omit Anglo-Saxon sources, of which there are an overwhelming number. The application sections - methods and results - are detailed and good. However, the "Conclusion" section is very brief and does not go into detail on the results. Then the crucial "Discussion" section is missing, which would confront the results with other relevant sources. A discussion of the limitations of the methods used is also absent. In the case of the use of the model area, there is no discussion of local specificities (e.g. including the topology of the transport networks).

Formally, I criticise the numerous typographical errors (e.g. missing spaces for brackets). The use of italics as notes in paragraphs is also unusual (e.g. for the chapter 'Related Work').

Overall, I recommend the article for a significant and serious revision.

Reviewer 2 Report

In this work, the authors consider the vehicles types and the spatial proximity of expressway, and a novel deep learning framework based on spatial-temporal attention mechanism is designed to capture the spatial-temporal correlation of travel time. At the same time, the experimental results verify the effectiveness of this work. The following comments are my concerns.

1. English needs significant improvement. All composition and grammar mistakes should be corrected. Note that even if the technical contents of a paper are acceptable, if the language is not acceptable, then the paper will not be published.

2. I suggest the authors rewrite the introduction since no works are cited. The introduction the section should contain the scope, significance of the research by summarizing current understanding and background information, stating the purpose of the work, and highlighting the potential outcomes.

3. Related work needs to be enhanced and supported by tables and more references are required to prove the importance of work. The authors should discuss the limitations of current related work and how they overcame these limitations in the proposed protocol.

4. The authors claimed “We conducted extensive experiments on several real-world traffic datasets. The results show that our method consistently outperforms the competing baselines.” in Introduction. However, only ETC data used.

Round 2

Reviewer 1 Report

The authors made many improvements and I think that this version is acceptable for publishing.

Reviewer 2 Report

In this revised version, I think the authors have answered my concerns. Hence, I recommend acceptance.